# The Role of Dielectrophoresis for Cancer Diagnosis and Prognosis

**DOI:** 10.3390/cancers14010198

**Published:** 2021-12-31

**Authors:** Giorgio Ivan Russo, Nicolò Musso, Alessandra Romano, Giuseppe Caruso, Salvatore Petralia, Luca Lanzanò, Giuseppe Broggi, Massimo Camarda

**Affiliations:** 1Urology Section, University of Catania, 95125 Catania, Italy; 2Department of Biomedical and Biotechnological Science (BIOMETEC), University of Catania, 95123 Catania, Italy; 3STLab s.r.l., Via Anapo 53, 95126 Catania, Italy; massimo.camarda@stlab.eu; 4Haematological Section, University of Catania, 95125 Catania, Italy; sandrinaromano@gmail.com; 5Department of Drug and Health Sciences, University of Catania, 95125 Catania, Italy; forgiuseppecaruso@gmail.com (G.C.); salvatore.petralia@unict.it (S.P.); 6Department of Physics and Astronomy “Ettore Majorana”, University of Catania, 95123 Catania, Italy; luca.lanzano@dfa.unict.it; 7Pathology Section, Department of Medical, Surgical Sciences and Advanced Technologies “G.F. Ingrassia”, University of Catania, 95123 Catania, Italy; giuseppe.broggi@gmail.com

**Keywords:** circulating tumor cells, dielectrophoresis, prostate cancer, detection, prognosis

## Abstract

**Simple Summary:**

Dielectrophoresis (DEP) is a label-free cell manipulation technique based on electrical differences that can be applied to liquid biopsies in the context of cancer prognosis and early diagnosis. In fact, it can overcome the current limitations of other platforms in detecting circulating tumor cells (CTC) in prostate cancer (PC), such as the low number of CTCs detected (less than one CTC/mL of peripheral blood), high-cost and time-consuming means for detection, quantification, isolation and characterization. The aim of this review is to give deep insights into the role of DEP for the diagnosis and prognosis of cancer.

**Abstract:**

Liquid biopsy is emerging as a potential diagnostic tool for prostate cancer (PC) prognosis and diagnosis. Unfortunately, most circulating tumor cells (CTC) technologies, such as AdnaTest or Cellsearch^®^, critically rely on the epithelial cell adhesion molecule (EpCAM) marker, limiting the possibility of detecting cancer stem-like cells (CSCs) and mesenchymal-like cells (EMT-CTCs) that are present during PC progression. In this context, dielectrophoresis (DEP) is an epCAM independent, label-free enrichment system that separates rare cells simply on the basis of their specific electrical properties. As compared to other technologies, DEP may represent a superior technique in terms of running costs, cell yield and specificity. However, because of its higher complexity, it still requires further technical as well as clinical development. DEP can be improved by the use of microfluid, nanostructured materials and fluoro-imaging to increase its potential applications. In the context of cancer, the usefulness of DEP lies in its capacity to detect CTCs in the bloodstream in their epithelial, mesenchymal, or epithelial–mesenchymal phenotype forms, which should be taken into account when choosing CTC enrichment and analysis methods for PC prognosis and diagnosis.

## 1. Introduction

Collection and analysis of tumour cells and tumour-derived products present in body fluids are referred to as liquid biopsy (LB). LB is becoming an important tool to complement conventional tissue biopsies for therapeutic decision-making in personalized treatment strategies. Several current strategies focus on the detection of circulating tumour DNA (ctDNA) in peripheral blood and the analysis of circulating tumour cells (CTCs), as cell-based LBs would provide complementary and clinically relevant information not only on DNA but also on proteins, RNA and cellular functions such as drug responsiveness. The incorporation of CTC-based LBs into standard diagnostic and treatment guidelines suffers from three fundamental limitations: (i) a low number of CTCs, (ii) genotypic and phenotypic heterogeneity, and (iii) high-cost and time-consuming means for detection, quantification, isolation and characterization.

To overcome these limitations, a promising but highly challenging approach is to massively augment the analyzed blood volume. In this context, diagnostic leukapheresis (DLA) has been established and subsequently validated [1] as a safe procedure to potentially screen up to 2.5 L of blood, thus increasing CTC yield up to 100-fold as compared to the volume of a standard blood sample (10 mL). Thanks to this higher sensitivity, via DLA, CTCs have been detected on larger numbers not only in metastatic patients but also in non-metastatic patients [1], even potentially allowing for the culturing of CTCs to obtain organoids, able to provide a platform for ex vivo treatment modelling [2].

The bottleneck of the DLA methodology, when used with current enriching technologies, is the high background of co-isolated white blood cells (WBCs), which limits the use of complete DLA products.

It is worth mentioning that there is no gold standard right now to detect and count CTCs, but each technique presents both advantages and disadvantages, while their combination may allow a more comprehensive and exhaustive characterization [3]. Most of the approaches use the expression of specific markers at the cellular surface level to separate CTCs from healthy cells [4]. The most widely used marker for the selection of CTCs is the epithelial cell adhesion molecule (EpCAM) [5]. A limitation of the EpCAM-based approaches is the fact that CTCs that have undergone the epithelial–mesenchymal transition (EMT) will not express EpCAM anymore, leading to an underestimation of the CTC number [6]. Alternative markers are the mucin 1(MUC1) and epidermal growth factor receptor (EGFR) [7]. Additional approaches include the use of microfluidic platforms (discussed in more detail later) as well as devices allowing for the online in vivo capturing of CTCs and the subsequent analysis by highly sensitive multiplex RT-qPCR assays [8]. Size-based approaches for the isolation of CTCs, including filtration methods, use, instead, physical properties of the different cells [9].

Although these have the potential to become valuable tools for cancer prognosis and diagnosis [10], the presence of CTCs in patients with a tumor is still strongly dependent on the platform used for detection, suggesting important limitations for currently available technologies. To date, only the CellSearch assay (Janssen Diagnostics, Raritan, NJ, USA) has received the USA Food and Drug Administration approval [11], while the AdnaTest Prostate Cancer (Qiagen, Hilden, Germany) is emerging thanks to its properties in immunomagnetic enrichment [12].

Among the strategies aiming at both detection and enrichment of CTCs, microfluidic and, more specifically, dielectrophoresis (DEP)-based technologies represent some of the most promising and attractive separation mechanisms.

DEP is a label-free cell-manipulation method that is based on electrical, rather than physical or biochemical, differences and may offer promising opportunities to improve CTC detection by separating rare cells simply on the basis of their specific electrical properties [13]. However, current DEP systems suffer from critical limitations, mainly associated with their low throughput, which has hindered their standardization as well as their widespread use.

Based on all these premises, the aim of the current review is to give deeper insights into the role and future perspectives of DEP in the context of CTC detection for cancer.

## 2. Dielectrophoresis: From Physics to Biological Applications

Dielectrophoresis refers to the ponderomotive movement of dielectric particles in inhomogeneous electric fields. The force on the particles, and thus their movement, is generated by the interaction of the externally applied electric field with the electrical dipole of the particles, which, in turn, is induced by the electric field itself (see Figure 1). Since the field is non-uniform, the pole experiencing the greatest electric field will dominate over the other, and the particle will move. Notice that this movement is independent of the direction of the background electric field. Thus, the particle will experience a movement also in the case of alternating electric fields, i.e., fields for which the electric potential of the external electrodes changes with time (compare Figure 1 top and bottom rows).

The direction of the dipole, and thus of the force, depends on the difference in polarizabilities, or permittivities, of the particles as compared to those of the suspending medium, which in turn depends on the frequency of the applied electric field, thus allowing for frequency-dependent manipulations of particles.

DEP allows for the controlled manipulation of micro- and nano-sized particles dispersed in colloidal solutions. Application fields include: cell partitioning and isolation [15,16], bio-structure assembling [17], nanostructure (e.g., carbon nanotube) deposition [18] or even filtration systems for oil purification [18]. DEP can thus be used to manipulate, transport, separate and sort different types of particles, typically in the nanometer and micrometer size. Particularly, since biological cells have cell-specific dielectric properties [19], DEP has many medical applications, and several experiments have been successfully made to separate cancer cells from healthy cells in peripheral blood samples.

For most DEP applications on cells, we can consider the simplifying case of spherical noncharged particles of radius R in conductive mediums for which the time-averaged force <*F_DEP_*> can be written as [19]:
$$F_{DEP}=2\pi R\varepsilon_{m}f_{CM}\nabla {\left|E_{rms}\right|}^{2}$$

where ∇*Erms* is the electric field gradient, *ε_m_* is the complex polarizability of the suspending medium, and *f_CM_* is the Clausius–Mossotti function [15,19,20], which incorporates the differences in polarizabilities of the particles and medium. This equation shows that, depending on the sign and module of *f_cm_*, the particle will move either in the direction of the field gradient (positive DEP, pDEP) or in the opposite direction (negative DEP, nDEP) (see Figure 1). *f_cm_* is in turn defined as:
fCM=Re[εp*−εm*εp*+2εm*]

where the terms 
$\varepsilon_{p}^{*}$
 and 
$\varepsilon_{m}^{*}$
 are the complex polarizability of the suspended object and the medium, respectively, generally defined as 
$\varepsilon_{p,m}^{*}=\varepsilon_{p,m}+\sigma_{p,m}/\left(j2\pi f\right)$
, with 
$\varepsilon_{p,m}$
 and 
$\sigma_{p,m}$
 representing the particles or medium permittivities and conductivities, *j* is the imaginary operator, and *f* is the frequency of the external applied electric field. From this equation, it can be seen that varying the frequency will change the *f_cm_* value and thus the DEP response.

In order to separate different cells, their complex permittivities, 
$\varepsilon_{p}^{*}$
, need to be identified, and the frequency of the applied electric field specifically chosen to achieve separation. For this, single- or multi-layered shell models are employed. The simplest case, sufficient for many applications, is the single-shell spherical model, which simplifies cells as spherical objects enclosed by an extremely thin cell membranes [19]:
$$\varepsilon_{p}^{*}=\varepsilon_{mem}^{*}\frac{{\left(\frac{r}{r-\delta }\right)}^{3}+2\left(\frac{\varepsilon_{cyt}^{*}-\varepsilon_{mem}^{*}}{\varepsilon_{cyt}^{*}+2\varepsilon_{mem}^{*}}\right)}{{\left(\frac{r}{r-\delta }\right)}^{3}-\left(\frac{\varepsilon_{cyt}^{*}-\varepsilon_{mem}^{*}}{\varepsilon_{cyt}^{*}+2\varepsilon_{mem}^{*}}\right)}$$

where *r* is the cell radius and 
δ
 is the membrane thickness, 
$\varepsilon_{cyt}^{*}$
 is the complex permittivity of the cell cytoplasm, and 
$\varepsilon_{mem}^{*}$
 represents the complex permittivity of the membrane.

Figure 2 shows typical *f_cm_* as a function of applied electric field frequency: for very low frequencies, *f_cm_* is negative, indicating a movement towards lower electric field regions, i.e., away from the electrodes, nDEP; for medium frequencies *f_cm_* is positive, pDEP; whereas, for very high frequencies, typically above 50 MHz, *f_cm_* turns negative again. As the frequency passes through well-defined crossover frequencies, *f_CO_*, fcm and thus the DEP force passes through zero and changes direction. Since different cells will have different *f_COs_*, it will be possible to separate them by applying electrical frequencies within the *f_CO_* of the different cells. As an example, referring to Figure 2, by applying a frequency of 100 kHz, the MDA cells will be subjected to a pDEP force, whereas white blood cells (WBC) will be subjected to nDEP, thus achieving separation.

The DEP crossover frequency is thus the essential parameter exploited for separating cells. In the case of a medium with conductivities much lower than the cytoplasm of the cells, *f_CO_*_1_ can be written as [21]:
$$f_{CO1}=\frac{\sigma_{m}}{R\varphi C_{0}}$$

where 
$\sigma_{m}$
 is the medium conductivity, *R* the outer cell radius and 
$\varphi C_{0}=C_{mem}$
 is capacitance per unit area of the cell membrane. 
$C_{0}$
 is the capacitance per unit area of the smooth plasma membrane, determined as 
$C_{0}\approx 0.009$
 F/m [22] and 
φ
 is the folding factor characterizing the different membrane features, such as ruffles, folds, and microvilli. 
Rφ
 can be considered as the “dielectric phenotype” of a given cell type determining its response to DEP manipulation. This different dielectric phenotype, together with cell size, gives DEP technology an important “separation capability” when compared with other separation techniques. Figure 3 shows the DEP responses of CTCs associated with different cancers as compared to those of healthy blood cells [23].

As already mentioned, 
$f_{CO1}$
 is sensitive to membrane features. This is because, at low frequency, the electromagnetic field is shielded by the low-conductivity membrane, making it more sensitive to cell shape, morphology and plasma membrane properties. When the frequency becomes higher, the electric field starts penetrating inside the cell, thus “probing” its content. For this reason, 
$f_{CO2}$
 can be used to discriminate cells based on cytoplasm features [25], though, because of the technical difficulties associated with the high-frequency actuation and the high sensitivity of the DEP response to small cell variations within cell subpopulations, the separation of cells based on 
$f_{CO2}$
, generally referred to as ultra-high-frequency, UHF, DEP, has not been extensively explored as compared with low-frequency, 
$f_{CO1}$
-based, DEP.

The dielectrophoretic force, and thus the separation mechanisms, increases with the square of the electric field. For this reason, high voltages are generally used. Unfortunately, under these conditions, cells can develop either field-enhanced ion losses (or electroporation at an even higher voltage [26,27]) or degrade due to direct heat. For this reason, to compromise between separation efficiency and cell deterioration, the applied voltage is generally in the range of some volts.

Another important factor that can impact the efficiency and purity of DEP separation is the cell loading concentration. This is because, in the presence of the electric field, nearby cells can interact through their induced dipoles [13,28]. This interaction can result in specific cells clustering, leading to lower device specificities (contamination of collected cells) [15]. Another complication arises from the particle–wall dipole interaction, i.e., the behavior of cells nearby the channel walls, which can result in a specific electrically induced adhesion, again resulting in lower device sensitivities (reduction of collected cells) [29].

To generate well-controlled non-uniform electric fields, typical DEP chambers use an array of interdigitated electrodes which, depending on the frequency and thus the *f_cm_* term, can lead to uniform repulsion from the electrodes (nDEP) located at the bottom of the channel or attraction towards them (pDEP) (Figure 4).

Currently, the most advanced DEP-based CTC separation system is the ApoStream^TM^ (Precision Medicine Group, LLC, Bethesda, MD, USA), based on a complex continuous field-flow DEP fractionation schema (FFF-DEP) [30].

The system has been successfully tested for different solid tumors to capture epithelial-like CTCs as well as mesenchymal- and stem-like CTCs [31]. The main limit of ApoStream^TM^, as for similar DEP-based sorting systems, is the low cell throughput, lower than 200 million (2^8^) cells/h. Furthermore, the technology is characterized by a complex multi-injection system (see Figure 5) that decreases its discriminating power. The limitation of low throughput is common to all current CTC-separating technologies, using physical, electrical or molecular separation schemes (see Table 1). This bottleneck limits the complete analysis of high-volume samples, resulting in lower sensitivities and ultimately impacting the capability of these technologies in assessing minimal residual diseases (MRS).

Since the advantages of DEP include the possibility to eliminate the labeling of the cells and take advantage of their dielectric properties, it has been suggested as a potential application for cancer stem cells (CSCs) detection. DEP can be applied for the detection of many sub-types of stem cells, including mesenchymal cells, neural stem cells, one marrow-derived mesenchymal stem cell, neural stem/progenitor cells and adipose tissue-derived stem cells [37]. However, for the sorting of CSCs, current DEP applications have been focused on glioblastoma stem cells (GBSCs) [25,38]. Interestingly, Lambert et al. suggested that the expression of biological GBSCs markers and the measurement of the ultra-high-frequency crossover frequency *f_x_*_02_ were closely linked [25].

$$f_{x02}=\frac{\sigma_{int}}{2\pi }\sqrt{\frac{1}{2\varepsilon_{m}^{2}-\varepsilon_{int}\varepsilon_{m}-\varepsilon_{int}^{2}}}$$


The authors demonstrated that at this frequency range, ultra-high-frequency DEP (above 50 MHz) was relevant in investigating the stemness status of cancer cells that exhibited undifferentiated features.

## 3. The Role of DEP for CTC Detection in Prostate Cancer

Prostate cancer (PC) represents the most common incident cancer in men in developed countries in 2021 [39]. Incident cases increased more for PC than any other malignancy globally, irrespective of development status.

In this context, liquid biopsy via isolation of CTCs represents a promising diagnostic tool capable of supplementing state-of-the-art PC diagnostics [40] and prognosis [41,42,43,44,45].

To date, only the CellSearch assay (Janssen Diagnostics, Raritan, NJ, USA) has received the USA Food and Drug Administration approval [11], while the AdnaTest Prostate Cancer detection kit (Qiagen, Hilden, Germany) is emerging thanks to its properties in immunomagnetic enrichment [12].

Another platform is the cytology-based filtration method that has the possibility to detect any tumour surface markers on cancer cells, epithelial and non-epithelial, and distinguish between single CTC and CTC clusters [46], whereas most other CTC technologies, such as AdnaTest or Cellsearch, critically rely on the epithelial cell adhesion molecule (*EpCAM*) marker, limiting their applicability for early-stage PC where CTCs may not have developed epithelial characteristics.

Similarly to other cancers, CTC in PC consists of different types of cells. When tumor cells are disseminated from the primary site, they require a transformation from epithelial cancerous cells to epithelial–mesenchymal transition (EMT). To grow into metastatic tumors, cancerous cells reverse back to their epithelial status through mesenchymal–epithelial transition (MET) [47,48]. These sequential transitions between EMT (dedifferentiation) and MET (differentiation) are powered by cell plasticity, which is an essential property of cancer stem cells (CSCs).

Currently, markers of PC-CSC populations include CD44/CD133, aldehyde dehydrogenase (ALDH), CD166 and CD44+/CD24− [49]. This phenotype can be detected with fluorescence-based cell sorting, fluorescence microscopy, in vitro and in vivo cell tracking, but they suffer from heterogeneous marker expression secondary to their self-renewal properties.

DEP is an *epCAM* independent, label-free enrichment system, separating rare cells simply on the basis of their specific electrical properties [13]. The technique is capable of isolating all types of solid tumour-related immortalized cells from the NCI-60 panel [24], and more recently to co-collect, together with CTCs, also stem-like cells (CSCs) and mesenchymal-like cells (EMT-CTCs) [50].

Depending on the frequency of the externally applied electric fields, DEP cell sorting (DACS) will be mainly dependent on membrane morphology (medium frequencies, 10–200 kHz) or cytoplasm composition (high frequencies, 5–400 MHz) rather than on a cell’s phenotype as for fluorescence cell sorting (FACS) or magnetic cell sorting (MACS). The most advanced DACS system, Apostream™ (Precision Medicine Group, LLC), is based on a complex continuous field-flow DEP fractionation schema (FFF-DEP) [51]. The system was successfully tested for different solid tumours to capture epithelial-like CTCs as well as mesenchymal- and stem-like CTCs [50,51]. The main limit of ApoStream™, as for similar DEP-based sorting systems, is the use of planar, interdigitated electrode configurations causing exponentially decaying DEP forces, which limits the size of the sorting volume on which the DEP forces can act efficiently and reduces cell throughput. In the case of ApoStream™, less than 200 million (2^8^) cells/h can be processed. Thus, a DLA sample would be processed in >10 h.

Le Du et al. [50] were able to demonstrate that ApoStream™ was successful in detecting EMT-CTCs among patients after neoadjuvant chemotherapy in breast cancer.

All patients who had at least one CTC had epithelial and/or EMT-CTCs; no patients had only CSC-CTCs. The detection rates of CSC-CTCs were 9% (4 of 47 samples), 22% (8 of 37 samples), and 19% (6 of 31 samples) at time points T0 (before chemotherapy), T1 (after chemotherapy but before surgery), and T2 (after surgery).

Although it has not been investigated in PC patients, ApoStream™ was able to detect CTC using laser capture cytometry in blood samples from cancer patients (NSCLC adenocarcinoma, breast cancer, ovarian cancer and squamous lung cancer patients) [51].

Based on all these premises, DACS advantages could also be translated in the early phase of cancer, when CTC can be potentially available.

It is important to state that the early screening in PC induces many undesired effects, including anxiety from false-positive PSA tests and complications from the further investigation with prostate biopsy, including hospitalization for infectious complications or rectal bleeding. One major problem of PSA screening is overdiagnosis, that is, the diagnosis of indolent, slow-growing prostate cancer that would otherwise not be diagnosed during the man’s lifetime. Indolent disease is typically defined in terms of cancer grade. Gleason score 6, also termed grade group 1, is low-grade cancer that does not require immediate treatment. High-grade disease, for which treatment should be considered, is defined as Gleason score 7 (grade group 2) or higher. Overdiagnosis turns healthy men into patients, which may take its toll on psychological well-being and quality of life. Most importantly, over the past two decades, most men with low-risk prostate cancer in the United States have undergone treatment with surgery and radiation. Such overtreatment has no, or almost no, benefit in terms of mortality reduction but leads to important and persistent side effects, most notably urinary and erectile dysfunction. Therefore, the correct diagnosis of prostate cancer is crucial to detect real positivity for the disease and avoid false positives due to current PSA accuracy.

In fact, although it is difficult to detect the EpCAM marker in early prostate cancer, an experience on the Adnatest platform was published, identifying CTCs even in patients with low-risk clinically localized prostate cancer (22.2%) and high-risk clinically localized prostate cancer (30.9%), showing that EpCAM CTCs, already with epithelial characteristics, can be detected in the context of early-stage PC [52].

In a study performed by Riedl et al. in 2020 [53], combining blood filtration and microscopic analysis using standard cyto-pathological criteria, the authors investigated the role of CTCs in a cohort of 47 patients with suspected PC.

The authors demonstrated the presence of CTCs in 25 out of 27 men in the early detection group. Of these, 20 were PSA-positive and were positive for PC. All these results led to 97% sensitivity and 99% specificity for PC by a mean CTC count of 3.1 per mL.

However, it is important to underline that the evolution of CTC proceeds via multiclonal expansion that causes the tumor to be composed of multiple cell subpopulations. The mechanism of metastases spread consists of several sequential steps: local invasion of the primary tumor cells, intravasation, extravasation, and the establishment of distant metastases (Figure 6).

During the phase of initial local invasion, substantial changes in the morphology of tumor cells occur. In the single-cell invasion pathway, epithelial cells undergo epithelial–mesenchymal transition (EMT), that is, the loss of epithelial characteristics and the gain of mesenchymal characteristics. Tumor cell size, including CTCs, ranges from 9 to 30 μm. Blood capillaries have a diameter of 3–8 μm, and thus CTCs can become trapped in their lumen [54].

Its application has found interesting results in the context of androgen-directed therapies via AR-V7 (androgen receptor-V7) signaling. Davis et al. evaluated ApoStream^TM^ technology to capture AR-V7 expressing CTCs from blood originating from primary tumor cells (epithelial) and those that have undergone epithelial–mesenchymal transition (EMT) of CRPC patients. They found that ApoStream^TM^-enriched cancer cells from cell lines expressed AR-V7 in both epithelial CTCs (detected in 6/10 patients in CD45− [median: 9.5] and in 7/10 CD45+ [median: 1] cells and mesenchymal CTCs (detected in 5/10 patients for both CD45− [median: 0.5] and CD45+ [median: 0.5] cells) [55].

Future possibilities could also include the use of urine as a source of CTCs. In fact, it is an obvious and natural choice for a biological fluid to yield diagnostically relevant amounts of cancer cells from patients with diverse urologic cancers, including PC [56]. Since the prostate gland is anatomically connected to the lower part of the urinary tract in males, diverse cells can be shed from the gland into the urinary tract, including tumour cells, which can be concomitantly washed out in the process of urination. The major advantage of tumour cell isolation via liquid biopsy of urine in comparison with the isolation of CTCs from the blood, as a truly non-invasive method, is a lack of limitation in the sample volume. Compared with blood, urine can be readily collected, stored, and transported. The urine collection process has enviable patient compliance and does not require a skilled medical professional.

DEP has been studied for the detection of voided cells in urine from bladder cancer patients. Statistically significant differences (*p* = 0.034) between the characteristics of the DEP spectrum of clinical samples have been reported, obtaining a sensitivity of 75% and specificity of 88%, in line with many molecular methods [57].

Finally, the utility of DEP is its capacity to detect cancer in the bloodstream in their epithelial, mesenchymal, or epithelial–mesenchymal phenotypes, which should be taken into account when choosing enrichment and analysis methods.

## 4. The Role of DEP for Breast Cancer

Breast cancer (BC) is one of the most common cancers in women, with an estimated 281,550 new cases in the United States [39].

Similarly, DEP has been partially investigated to detect CTC in breast cancer, but most of the studies were pre-clinical.

Maltoni et al. isolated and studied CTCs in the peripheral blood (PB) of 48 EBC patients pre-surgery and one at six months post-surgery using an approach involving EpCAM-independent enrichment and a dielectrophoresis-based device [58]. His approach allowed us to detect three phenotypic classes: EpCAM-positive/panCK-positive, EpCAM-negative/panCK-positive and EpCAM-positive/panCK-negative. Although it was not possible to make further distinctions between these three classes, our method did detect a more general “epithelial” phenotype. Conversely, the widely used CELLSEARCH^®^ system only enriches and detects ‘double positive’ CTCs, i.e., EpCAM-positive/panCK-positive CTCs.

DEP has also been used to measure membrane potential as an additional biophysical parameter to be considered in the assessment of the cell line panel. In fact, DEP has been demonstrated to be useful to study multi-drug resistant (MDR) breast cancer cells. Coley et al. revealed that cytoplasmic conductivity was affected by the movement of molecules and that altered membrane potential was associated with an MDR phenotype but in a complex manner. DEP data suggest a model whereby relative increases in cytoplasmic conductivity were correlated with MDR, whilst relative decreases equate with a sensitized phenotype, e.g., MCF-7TaxR. In particular, the extent of anthracycline accumulation was inversely related to cytoplasmic conductivity [59].

An et al. analyzed the parameters that influenced DEP separation. In particular, the authors separated MCF 7 and MCF 10A cells efficiently under an applied voltage of 48 MHz–8 Vpp and a flow rate of 290 μm/s using the advanced DACS devices. MCF 7 and MCF 10A cells were separated with a maximum efficiency of 86.67% and 98.73%, respectively [60]. In another report, MCF57 cells responded at a high frequency between 850 kHz and 20 MHz [61].

## 5. Microfluidic Technologies for CTC Isolation and Analysis

One of the main aims of biological and biochemical studies is to understand the mechanisms characterizing a single cell, which is the minimal functional unit of life. In this context, the application of microfluidic-based devices offers numerous advantages, including the rapid and simultaneous separation and quantification of multiple chemical species, high sensitivity and reproducibility, and the possibility to study the heterogeneity of cell populations [62], the last being an extremely important factor to be considered for the analysis of CTCs. The use of these devices is also accompanied by a very short analysis time and gives the opportunity to use different detection platforms, such as fluorescence and electrochemical detection, just to name two, making it useful for high-throughput single-cell analysis. All these features make microfluidic-based technologies a powerful analytical tool for biomarker detection in personalized therapy and precision medicine.

Over the past two decades, the enormous potential of microfluidic-based technologies for isolating and detecting CTCs from whole blood has emerged. As described in a very recent review written by Cheng and co-workers [63], thanks to the advancement of microfabrication and nanomaterials, different approaches have been developed for isolation (capture + release) and analysis (morphology, genomics, and protein)/profiling (transcriptomic and functional) of CTCs on microfluidic platforms; these new approaches are characterized by numerous benefits, especially in terms of cell-capture efficiency, purity, detection sensitivity, and specificity, allowing to better explore cancer mechanisms and address increasingly complex biological questions. Using this as a very good example of microfluidic applied to CTC analysis, a fast and efficient microfluidic cell filter for label-free isolation of CTCs from unprocessed peripheral blood obtained from colorectal cancer patients has been developed by Ribeiro-Samy et al. [64]. This device (named CROSS chip) made it possible to capture CTCs contained in 7.5 mL of whole blood based on their size and deformability in a short time with high purity and efficiency. CTC enumeration by CROSS chip allowed the stratification of patients with different prognoses. The cells isolated using this microfluidic device were lysed and further subjected to molecular analysis by using droplet digital PCR, revealing a mutation in the adenomatous polyposis coli gene for most patient samples considered, confirming their colorectal origin and underlining the adaptability of this technology for downstream applications. Among the different technologies that can be coupled to microfluidic, DEP being label-free, fast, and accurate is very promising. In fact, as proven by numerous and often recent scientific publications, DEP is becoming a commonly used technique in microfluidic (DEP-on-a-Chip) for particles or cell separation and has been widely applied for bio-molecular diagnostics as well as for medical and polymer research.

## 6. DEP Applied to Microfluidic Platforms: Focus on Cell Separation and Analysis

As previously mentioned, DEP is becoming one of the most promising separation techniques for micro- and nanoscale systems thanks to its low running cost, rapid sample processing, and the possibility to be easily integrated into microfluidic devices [65]. These features, along with the high efficiency, sensitivity, and selectivity that characterize DEP, make it really attractive for the study of cell behavior, especially at the single-cell level. DEP has been used in combination with lateral field-flow fractionation (LFFF) to improve the isolation of spiked breast cancer cells from “healthy” blood cells [66]. In this research study, Waheed et al. developed a continuous-flow, DPE-LFFF microdevice able to isolate green fluorescent protein-labelled MDA-MB-231 breast cancer cells from regular blood cells. In a different study carried out by Piacentini et al., a device able to separate platelets from other blood cells by DEP-FFF was developed [67]. This innovative device uses the so-called “liquid electrodes” design (planar electrodes patterned at the bottom of dead-end chambers positioned perpendicularly to the main channel) and can be employed with low-applied voltages, giving a very efficient separation coupled to a very high purity of platelets (∼99%) with almost absent cell loss (<2%). This device is already set up for integration with an on-chip cell counter, allowing the measurement of platelet concentration in the blood. More recently, De Luca and co-workers developed a microfluidic platform combining DEP and imaging (DEPArray) that has been used to accurately select single breast cancer cells [68]. A hydrodynamic and direct-current insulator-based DEP (H-DC-iDEP) microfluidic device made of polydimethylsiloxane (PDMS) covered with a glass cover-slide allowing the separation of plasma from fresh blood has also been developed [69]. This represents the first device, making it possible for real-time monitoring of the plasma components without pre- or post-processing steps. As recently demonstrated, DEP microfluidic devices can be used for the separation of different types of cells, such as leukocytes (floating cells) and spermatozoa (moving cells) [70]. In particular, the differences in size and membrane properties were considered for the separation of leukocytes, while membrane charge and cytoplasm conductivity are the two key factors for the separation of X and Y spermatozoa. DEP coupled to microfluidics has also been used for the high-throughput selective capture of single CTCs [71].

Lastly, microfluidic can be applied for CTC detection in urine, which has been used as an alternative to blood for liquid biopsy as a truly non-invasive, patient-friendly test. Rzhevskiy et al developed a spiral microfluidic chip capable of isolating PC cells from the urine of PC patients. The model has been investigated in DU-145 cells, and authors reported >85 (±6)% efficiency. The microchannel was functional in at least 79% of cases for capturing GPC1+ putative tumour cells from the urine of patients with localised PCa [56].

## 7. The Use of Nanostructured Materials in DEP for CTC Detection

The nanostructured materials, thanks to their particular optical, electronic, photothermal, magnetic and chemical properties that are largely influenced by their quantum size effect, have attracted great interest in the various areas of the scientific community. Nanomaterials such as nanoparticles, nanorods, nanotubes, nanowalls and nanostructured coatings are perceived as the most promising nanotechnologies in the vast field of biomolecule detection, including nucleic acid, protein, and cell recognition [72,73], but their use in DEP for CTCs detection is still limited and leaves plenty of room for improvements.

Recent works have reported the use of nanostructured materials for CTC detection, using various nanotechnologies such as magnetic separation, nanofiltration and NanoVelcro. In the magnetic separation approach, CTCs interact with specific probes anchored on the surface of magnetic nanoparticles and are recognized on a microarray format [74]. NanoVelcro is a new technology recently developed by a research team at UCLA. It is based on silicon nanowire substrates coated with polymeric nanostructures such as polydimethylsiloxane [75], thermo-responsive Poly(N-isopropylacrylamide) [76] and conductive boronic acid-grafted nanocoating [77]. These nanostructures are properly designed and tested for efficient CTC isolation, purification and enumeration in a miniaturized NanoVelcro chip [78]. Nanostructured materials have been largely proposed as promising agents to increase the specificity for CTC recognition through chemical functionalization with molecular or biomolecular probes properly designed to interact with specific targets in the CTC membrane [79]. In this scenario, EpCAM antigen is one of the most used biomolecular targets widely expressed on the surface of CTCs [80]. This approach is used in the CellSearch system, a commercially available platform for CTC detection [81]. Moreover, carbon-based nanostructures such as graphene oxide, carbon dots and carbon nanotubes have been proposed as promising agents for CTC detection upon functionalization with EpCAM antibodies for direct detection of cancer cells in whole blood by electrical impedance sensing [82]. Moreover, nanostructured coatings have been reported to enhance interactions between substrates and targeted cell surfaces, with a net increase in cell affinity compared with flat substrates [83]. All these data have encouraged research teams to focus their efforts on the development of nanostructured materials for the detection of CTCs by DEP technology [84]. Wu and co-workers recently reported the fabrication and testing of an optical DEP device based on nanostructured PDMS coating, demonstrating an increase in CTC recognition performance [85]. Barik and co-workers developed a graphene-based DEP platform produced by a nanofabrication process, capable of reversibly trapping nanosized particles and biomolecules with nanoscale precision [83]. Swardy and co-workers developed a DEP microfluidic device based on silica beads modified with an antigen probe capable of binding single cells [86]. Cao et al. developed an iDEP platform comprising a structure with SiO_2_ microelectrodes coated with nanosized (100 nm) Ag-nanorods, they demonstrated an increase in the cell enrichment factor nearly ten times greater than the naked electrodes [87].

The above-mentioned nanomaterials, together with additional i nanotechnologies, will be further developed to obtain high sensitivity and a specific CTC detection method based on the DEP approach.

## 8. Monitoring Cell Integrity during DEP by Fluorescence Imaging

An important issue in cell-based DEP is whether the integrity of the cells is preserved during the process. Ideally, the biological, biochemical and biophysical properties of the cells must remain unaltered during DEP. In practice, cells are subjected to forces, and cell damage can occur for several reasons, including excessive charging of the cell membrane in the electric field, suspension in a non-physiological medium, and flow-induced shear stress [88]. The extent of cellular damage will be dependent on the conditions of device operations such as applied voltage, buffer, and flow rate, as well as biological parameters such as cell type. Thus, for any specific application, a careful optimization of the DEP protocols is required to preserve cellular integrity.

Cell viability is the most straightforward parameter to monitor and quantify the extent of cellular damage induced by DEP. Cell viability during the exposure to the DEP electric field (1 MHz, 10 V) has been quantified using the fluorescence of propidium iodide (PI), a cell-impermeant fluorophore that stains only dead or dying cells due to their loss of membrane integrity [89]. The cell viability of Jurkat cells was dependent on the exposure time and the size of the electrode [89]. However, under this condition, cell death was below 10%.

A more subtle question is which biochemical and biophysical alterations are induced in the cells during DEP exposure time and if they are relevant for any subsequent analysis performed on the cells. In this respect, we believe that an important role can be played by the application of fluorescence imaging approaches to characterize the biochemical and biophysical changes, if any, occurring during DEP.

Fluorescence labeling provides molecular specificity and allows direct mapping of the biochemical content of a cell. The resolution of currently available fluorescence imaging techniques ranges from the size of small organelles to that of single molecules. Imaging flow cytometry is an established technique that combines imaging with a typical resolution of 500 nm with the processing of thousands of cells per second [90] and is the technique of choice for rare cell detection. Confocal microscopy provides a higher spatial resolution, typically 200 nm in the lateral direction and 500 nm in the axial direction [91], sufficient to clearly visualize most of the subcellular components, but the typical number of cells analyzed is much lower, in the order of one cell per second. The recently developed super-resolution microscopy techniques provide spatial resolution down to 20 nm [91], enabling visualization of the finest molecular details. Finally, Förster resonance energy transfer (FRET) can detect biochemical interactions and molecular distances at a spatial scale below 10 nm [92].

Biophysical properties that can be monitored by fluorescence imaging include cellular and organelle shape, viscosity, and macromolecular architecture. A fundamental advantage of fluorescence is the capability of labeling multiple species with different colors. This allows colocalization analysis and the measuring of distances between molecular components [93]. For instance, multicolor imaging could be performed to simultaneously visualize different subcellular components and monitor their integrity during DEP. Another interesting property of fluorescence is the sensitivity of specific probes to the molecular environment. Environment-sensitive membrane probes have been used to monitor biophysical changes of cellular membranes [94]. These techniques could be applied to monitor biophysical changes of the plasma membrane that occur during DEP, eventually leading to more optimized DEP protocols.

## 9. The Role of CTCs in Precision Medicine

Cancer evolution and recurrence depend on the synergic interaction of molecular features (namely genomic mutations, single nucleotide polymorphisms, CpG island methylation) and phenotypic features of individual clones that proliferate out of physiological constraints, destroying tissue barriers to spread to other organs and promote immune evasion [95]. Spatial and temporal cancer heterogeneity arises from subclonal evolution, driven by the simultaneous presence of different mutational patterns that are the consequence of complex and altered molecular pathways that are potentially different for each individual patient and even between the same patient at different moments in the development of the disease [95].

A potential strategy to prevent cancer metastasis and provide clinical benefits to patients is the early detection of potential metastatic clones, carrying driver mutations, which are capable of leading to the development and guidance of the tumor phenotype, conferring a selective growth advantage to the cell. Driver mutations should be distinguished by passenger mutations that are accessory and do not play an active role in conferring clonal advantage [96]. The accumulation of driver and passenger mutations is not constant in all tumor cells, leading to a different growth rate of different subclones within the same tumor, characterized by different gene expression patterns, associated with different prognoses [95,96]. During biopsy or surgical resection (both late and invasive techniques), it is almost impossible to select only cancer cells because a mixed cellular representation between normal and tumor cells is present. There is increasing evidence of the intra-tumor heterogeneity in cancer due to spatial [97] and temporal heterogeneity [98], with a plethora of sub-clonal mutations, carried only by a fraction of the tumor cells [99]. For this reason, liquid biopsy, based on the detection in the bloodstream of circulating tumor DNA (ctDNA, tumor-derived fragmented DNA not associated with cells) and broader circulating free DNA (cfDNA, degraded DNA fragments released by apoptotic cells and necrotic cells, not necessarily of tumor origin) are emerging means for a non-invasive investigation of the tumor molecular structure [100,101].

Next-generation sequencing (NGS) can identify mutations even with a low representation (up to 3%) by means of very high coverage sequencing, but additional bioinformatics and machine learning tools are required to identify clinically relevant mutations in a background of errors, noise, and random mutations [102], challenging the contribution of passenger mutations that can still be used to improve cancer subclassification [101,103]. Because of genome plasticity, cfDNA might not be fully informed of cancer evolution, representing an average of multiple subclones present in the individual patient, providing the rationale for the search for alternative non-invasive and not-expensive platforms. In this context, microfluidic single-cell manipulation [104] for enumeration and isolation of CTCs is by far the best biological matrices to move from single gene analysis to single-cell profiling that is required for the new era of precision medicine [104,105,106].

Apart from the advantages of DEP in detecting CTC, this technology could be able to detect diverse sources of biomarkers for a wide variety of diseases. For example, a novel insulator-based dielectrophoretic (iDEP) device predicated on an array of borosilicate micropipettes was applied to isolate exosomes from conditioned cell culture media and biofluids, such as plasma, serum, and saliva. The device was capable of exosome isolation from small sample volumes of 200 μL within 20 min under a relatively low (10 V cm^−1^) direct current (DC).

CTCs are released into body fluids from primary or metastatic tumour sites for several, not always recognized, reasons, such as independence from adhesion to the supportive niche [107,108]. There are some technical issues to overcome to use CTCs isolated by a screening tool for early detection of solid cancer or its recurrence after treatment, including:-The sample sources. Based on CTC source (peripheral blood, urine, saliva, or other biological fluids), the spatial clonal heterogeneity could lead to false-negative results by underestimating the disease burden due to the presence of remaining tumor cells in not accessible sites, which could be monitored with coupled imaging techniques, such as PET or MRI or circulating free DNA (cfDNA) by NGS., Several means to isolate and manipulate CTCs have recently emerged, ranging from using microfluidic to dielectrophoresis techniques [109,110], starting from different kinds of biological fluids [111] other than peripheral blood. For PC, there is an emerging interest for seminal plasma since the electrophoresis of seminal plasma cfDNA can discriminate between subjects carrying tumor or benign proliferation [111]. Alternatively, tumor isolation via liquid biopsy of the urine lacks limitations in the sample volume. For PC, the first stream of urine (about 30 mL) is sufficient to collect most cells of interest using a spiral microfluidic chip. This approach requires urine filtration or a specific pipeline of preanalytical enrichment to discard large waste elements such as urine crystals [112].-The absolute number of CTCs recovered. On average, 5–50 CTCs can be recovered from every 7.5 mL of peripheral blood from a patient with metastatic cancer, meaning that a 10-5-10-6 sensitivity is required, which is the threshold commonly accepted in the evaluation of minimal residual disease in hematological cancers. CTC enumeration is clinically relevant since it is associated with high tumor burden, aggressive disease, and inferior progression-free survival [107,108,113]. However, tag-based techniques for CTC enumeration and isolation can underestimate the amount of CTCs due to the loss of cells without epithelial markers [114].-Post-CTC recovery processing to detect either phenotype such as next-generation multidimensional flow cytometry, imaging, transcriptomics, metabolomics and proteomics, or genotype aberrations (such as ASO-RQ PCR, digital droplet PCR, and NGS that can reach 10-6 sensitivity, and their standardization among different laboratories, to investigate, at the single-cell level, novel biological mechanisms associated with cancer metastasis and tissue homing [115,116,117,118,119]. The limited number of recovered cells can make impossible post isolation manipulation, requiring large amounts of biological fluids.-How to improve precision medicine. Since CTCs could show a unique morphology and profile of drug sensitivity different from the in-site tumor, which could be challenged to prevent tumor recurrence, raising the question of whether a comparison with the tumor in the primary sites or interactions with the immune system should be further investigated to predict cancer evolution dynamics. For example, in multiple myeloma, compared to primary tissue, CTCs are mostly quiescent (arrested in the subG0–G1 phase of the cell cycle) [108,117], by mirroring the entire heterogeneity of the tumor. For these reasons, it is crucial to test the optimal time points for CTC detection.

## 10. Conclusions

There is currently no technology available to detect CTCs in all their phenotype and dynamic processes. However, DEP seems promising by combining electrically based and label-free cell-separation platforms. The technology can be used for:(i)the detection, isolation and quantification of label-free CTC with up to one order of magnitude compared to current technologies once limits in throughput will be properly addressed;(ii)the detection, isolation, characterization, and quantification of adequate numbers of CTCs for early detection of tumor relapse and potential improvements of clinical application of LBs;(iii)the reduction of expensive, time-consuming and operator-dependent procedures, using a label-free technology;(iv)collecting of CTCs after DEP for functional assays such as drug testing or for the generation of in-vivo models (organoids, CTC-derived xenotransplant mice);(v)the analysis of tumor phenotype where no suitable antibodies are available to detect and catch CTCs such as prostate cancer due to the absence of known targetable surface proteins.

All these benefits, once adequately established on a clinical basis, will largely outweigh the main drawbacks of other platforms, achieving a long-lasting transformative positive effect, both on social and economic levels and on cancer therapeutic and screening strategies.

## Figures and Tables

**Figure 1 cancers-14-00198-f001:**
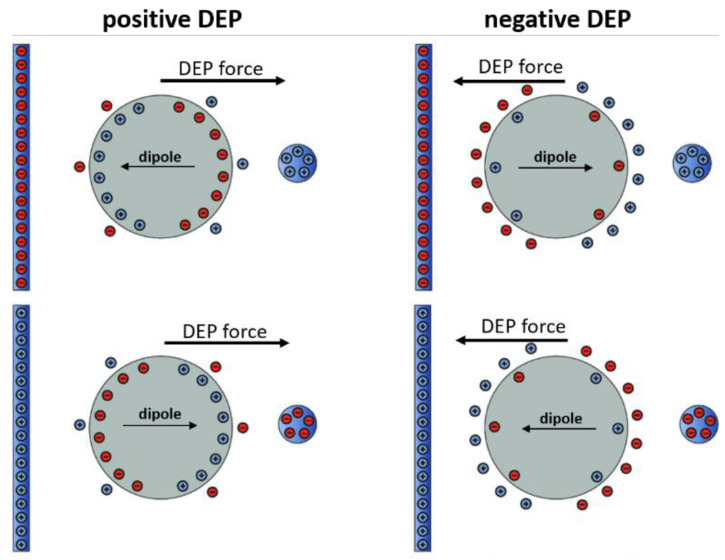
Polarization, induced net dipole and dielectrophoretic force direction for a particle that is (**left**) more polarizable than the surrounding medium and (**right**) less polarizable. Top and bottom are two opposite arrangements of the background electric field (adapted from ref. [14]).

**Figure 2 cancers-14-00198-f002:**
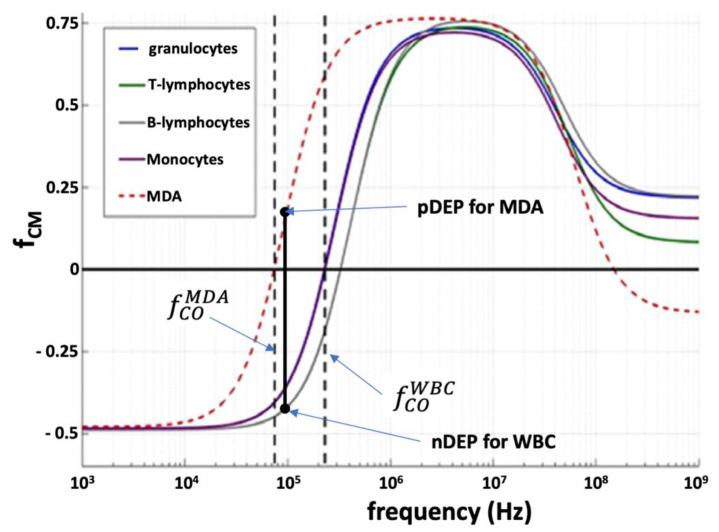
Clauss−Mosotti function for MDA, a CTC derived from breast cancer, as compared to healthy cell types in pheripheral blood.

**Figure 3 cancers-14-00198-f003:**
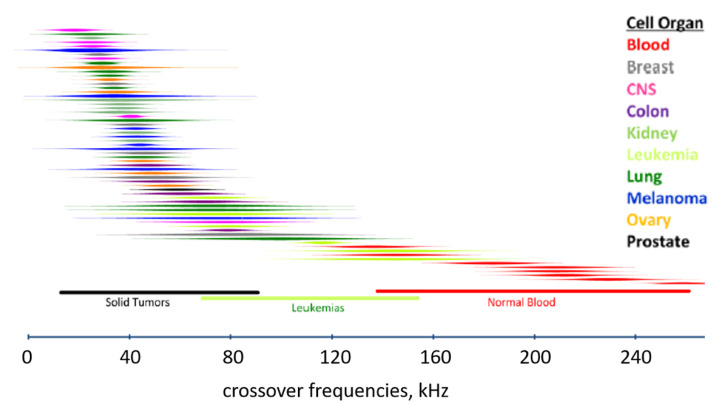
The DEP responses of cancer and normal blood cells expressed in terms of first crossover frequency 
$f_{CO1}$
 (Adapted from ref [24]).

**Figure 4 cancers-14-00198-f004:**
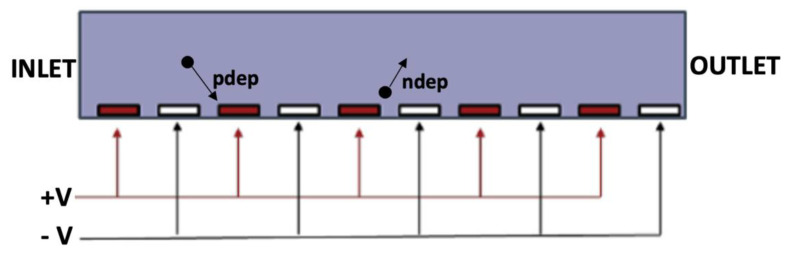
Side view of an interdigitated electrode channel, with indication of particle response depending on the sign of the DEP force.

**Figure 5 cancers-14-00198-f005:**
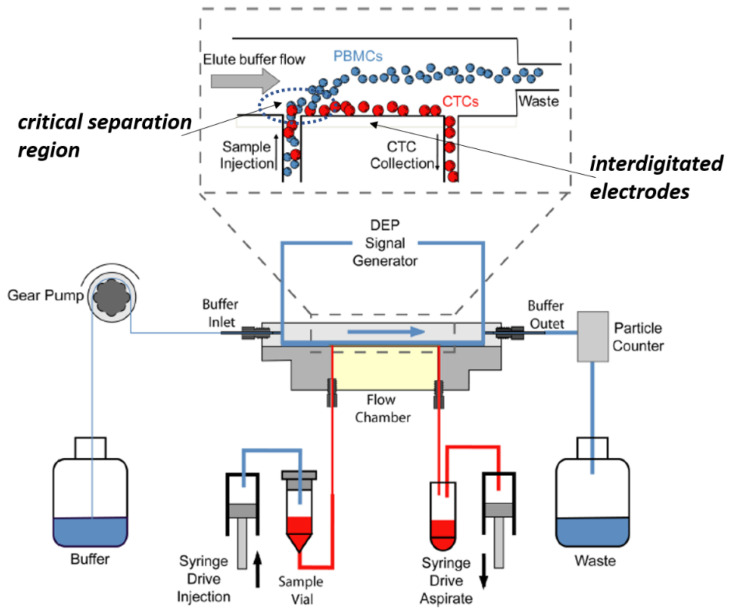
Schematic, side-view, of the ApoStream^TM^ device, showing the complex multi-inlet, multi-outlet, and microfluidic system. (Adapted from [30]).

**Figure 6 cancers-14-00198-f006:**
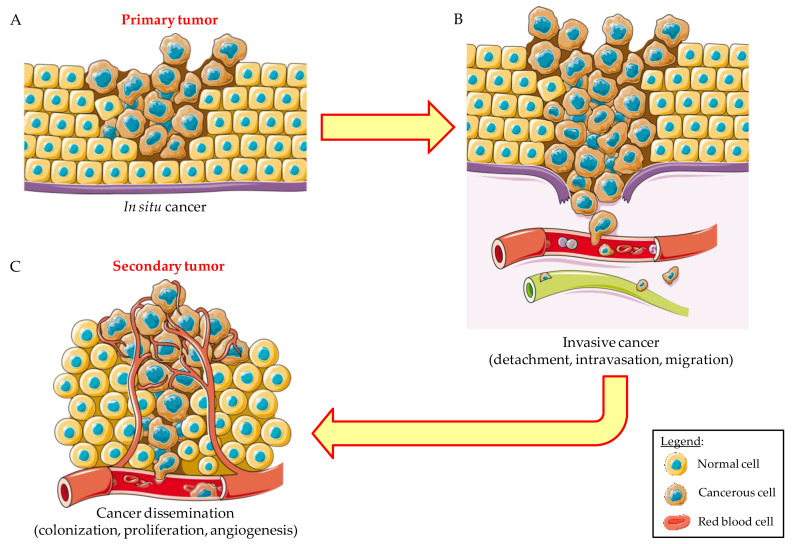
Schematic representation of the metastatic process. Metastasis is a multi-stage process starting with the formation and growth in situ of the primary tumor (**A**). Some tumors can become invasive due to the detachment of cells (CTCs) that are able to enter blood or lymph vessels, a process known as intravasation, and migrate to sites far from the starting point (**B**). The following step is the dissemination of cancer occurring when “traveling cells” extravasate (exit blood or lymph vessels) and colonize new sites forming secondary tumors (**C**). Once in the new sites, cancerous cells are able to proliferate and recruit the blood vessels (angiogenesis) needed for trophic support. (Created with https://smart.servier.com, accessed on 15 December 2021).

**Table 1 cancers-14-00198-t001:** Details of extracted device characteristics can be found in ref [32,33]. * Throughputs have been extrapolated based on processing time.

Features	CellSearch^®^	ISET^®^	^LP^CTC-iChip [34]	Cytofluorometers	ApoStream^TM^
** Throughput [Cells/h] **	**1.6 × 10^6^ ***	**<1 × 10^7^ ***	**≈1.5 × 10^9^ ***	**10^8^**	**<2 × 10^8^**
** * epCAM * ** ** + independent **	**NO**	**YES**	**YES**	**DEPENDS**	**YES**
** antibody independent **	**NO**	**YES**	**NO**	**NO**	**YES**
** Device specificity **	**HIGH**	**MEDIUM [35]**	**HIGH**	**HIGH**	**HIGH [36]**
** Downstream analysis **	**NO**	**YES**	**YES**	**YES**	**YES**
** Enumeration capability **	**YES**	**NO**	**NO**	**YES**	**YES**

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
