# Peer review of "The Role of Dielectrophoresis for Cancer Diagnosis and Prognosis"

_cancers, 2021, doi:10.3390/cancers14010198_

Round 1
Reviewer 1 Report
Russo et al. summarized the use of dielectrophoresis/DEP for detecting circulating tumor cells for prostate cancer. It is generally a field of great interest relevant to the journal's scope. This mini-review captured the important core aspects to promote the understanding of DEP and CTC detection. Minor comments are listed below with aims to improve clarity:
- The formatting needs to be double-checked, for example: "10,000 peripheral blood cells per mm3 Error! Bookmark not defined... A"
- The aspect ratios of the figures need to be reconsidered to fit the layout. The resolution of most figures is also insufficient and needs to be improved.
- The introduction section needs to be clearer for general readers outside of the DEP field. For example, the explanation of pDEP and nDEP in Fig. 1 could be improved. The fco1/co2 and the 4-star red points in Fig. 3 also need to be better annotated and explained in both the manuscript and the figure caption, as it is important for understanding Fig. 4. Nevertheless, what is missing is how these parameters help DEP stand out in differentiating CSCs from circulating tumor cells?
- The core argument of using DEP over other CTC detection/capturing technology, according to the author, is the ability to isolate cells independent of biological markers that is advantageous for stem-like cells (CSCs). However, throughout, the review article did not mention or review any relevant literature or data that support the potential use of DEP for CSCs. The differences between CSCs and general circulating tumor cells under DEP were also not clear? The entire review focus on circulating tumor cells hence the rationale behind the arguments needs to be carefully reconsidered.
- As the title suggests the use of DEP for prostate cancer, the content cited did not fully match the claim, i.e., Fig. 4 was the only DEP literature addressing the use of DEP for prostate cancer. Fig 3b was about breast cancer cells (cell lines, actually). The author should consider strengthening their focus and providing more relevant information that helps justify the leading edge of using DEP for prostate cancer biopsy.
Author Response
We would like to thank you for the suggestions that you have provided. Please find attached our answers that we believe will improve our manuscript. I would finally like to thank you in advance for your interest in our article, and we hope it will be suitable for publication in your prestigious journal.
- We have entirely revised the manuscript by correcting typos and mistakes.
- We have provided to change figures and improve the overall quality.
- We have modified the introduction regarding DEP. Figure 3 has been improved as per your suggestions and we have also improved this section regarding the relationship between DEP and CSC.
- We agree with this interesting observation. We have improved each section by inserting a discussion about the role of CSC detection. We have improved the manuscript by considering its potential role in the early prostate cancer
Reviewer 2 Report
The authors present a review paper entitled “ Challenging the Current Paradigm of Liquid Biopsy through Dielectrophoresis (DEP) in Prostate Cancer”. As a review paper, it is not necessary to review the quality of new science. Instead it is necessary to establish how well the paper fits the brief.
I have several issues with this paper. We can take the title as a starting point. "Challenging the Current Paradigm of Liquid Biopsy through Dielectrophoresis (DEP) in Prostate Cancer", is firstly ambiguous - are the authors challenging the paradigm of CTC detection or the paradigm of DEP-actuated CTC detection? The "liquid biopsy" part is a strange thing to highlight since the paper is actually about CTC detection, which would omit detection of early stage PC. Would DEP be able to identify cell differences in urine, for example? Would has already been published on detection of bladder cancer by DEP from urine samples - another form of liquid biopsy that does not involve CTCs. In fact, the paper has very little to do with PC specifically - it reads as a general DEP review of CTC detection. Prostate cancer amounts to 2 pages of specific content (half a page of which is a figure), and is ignored in the rest of the paper.
It is also not clear how the authors believe they are "challenging the paradigm". DEP is not used for PC detection - this is true but the paper doesn't really discuss PC detection, only CTC detection with a brief diversion to compare against CTC detection for PC on the two pages mentioned previously. As to challenging orthodoxy, the ApoStream platform described on page 6-9 is presented as if still in development; it has been commercially available for several years, provided as a service rather than as apparatus for sale. The service is available here: https://www.precisionformedicine.com/specialty-lab-services/tissue-liquid-biopsy/apostream-ctc-and-rare-cell-isolation-for-liquid-biopsy/.
Furthermore, the standard of English is very poor throughout. For example, one the first page sentence two reads: “ In fact, it can overcome current limitation of other platfmorm like low number of CTCs detected”. This should be “If fact,it can overcome the current limitations on other platforms, such as the threshold number of CTCs required”. Note that “platfmorm” is a typo. The abstract also mis-spells “technollgies” and “tontext”. This continues throughout the document; mistakes are not limited to English spelling and grammar, with the authors mis-spelling Clausius in the figure legend of Figure 3, for example. There is a classic “Error! Bookmark not defined” error on page 6. The text is riddled with errors that suggest a lack of careful reading before submission.
Taken together, I cannot recommend this paper for publication in its present form. This is not to say that there is nothing here that is publishable, but the authors must give the work a substantial overhaul before submitting a new version. I would suggest the following three mandatory changes:
- Acknowledge that this is a review about CTC detection, not PC detection. If it is about PC detection, then the paper should be about that and not PC specifically. If it is about PC specifically the authors need to address earlier stage detection (before metastasis) and consider other liquid biopsies, not just blood (given most PC detection biomarkers being developed are urine-based this should not be hard). If it's about CTC that's fine too but the authors should ay so.
- Acknowledge the progress that has been made in these areas and the place of this paper in relation to that progress. As I say, there is a commercial DEP service that has been available for several years. They did indeed challenge a paradigm but in the process set a new one. Is this paper aiming to challenge what Apocell (and now Precision for Medicine) have done, or to just expand on it? Do not overpromise.
- Proofread the paper thoroughly.
Author Response
We would like to thank you for the suggestions that you have provided. Please find attached our answers that we believe will improve our manuscript. I would finally like to thank you in advance for your interest in our article, and we hope it will be suitable for publication in your prestigious journal.
- We agree with your consideration. We have totally changed the title focusing at this time on the role of DEP for cancer detection and prognosis.
- We agree with this consideration. Our paper has the intention to improve current knowledge of DEP, putting together current status but also combine with future perspective. We strongly believe that DEP technology can be useful for CTC detection thanks to its label free sorting.
- We have corrected the manuscript with an English native speaker.
Round 2
Reviewer 2 Report
I suggest that the authors' efforts have certainly improved the paper. Whilst there are still errors, it is nearer publishable form than it was. Moving away from the ambiguous focus on prostate cancer is definitely an improvement, though I would consider either removing the PC section (section 3) and merging the references into the other sections, or consider creating other sections for other tumours and tumour types.
My biggest remaining concern is still that of focus I think the paper is better for changing emphasis from "detection of prostate cancer by DEP" to "the detection of cancer by DEP". This is a noble and achievable aim. However, as structured, the text of the paper remains focussed almost exclusively on the detection of CTCs. There are a number of different liquid biopsies that DEP may be applicable to - urine, saliva and others - and this is before we consider disaggregated cells. However, despite the fact that such work exists, it doesn't appear in the paper - or where it does (e.g. reference 57), it is mislabelled as detection of CTCs instead of analysis of cells in urine. I suggest that the authors need to go back to the text and references, and think about what the paper would look like if it were to properly serve the title - which is a worthwhile title and a paper the community would fine very useful. It needs more than CTCs, and it needs to be written with an acknowledgement that CTCs are not the only way (or even the best way) of using DEP to detect cancer. Either that, or the authors should be explicit that the paper is about DEP to detect CTCs, though I think the narrowing of scope would be a shame.
Author Response
We would like to thank again the reviewer for his precious comments.
as concerning the role of DEP in other cancer, we have added a section regarding breast cancer. Concerning other tumor like lung or colon the evidences were not such robust and we have preferred to omit. As concerning the use of DEP in other fluids, we added a paragraph regarding the role of iDEP in saliva.